# Closing the Dequantization Gap: PixelCNN as a Single-Layer Flow

**Didrik Nielsen**
Technical University of Denmark
didni@dtu.dk

**Ole Winther**
Technical University of Denmark
olwi@dtu.dk

## Abstract

Flow models have recently made great progress at modeling ordinal discrete data such as images and audio. Due to the continuous nature of flow models, dequantization is typically applied when using them for such discrete data, resulting in lower bound estimates of the likelihood. In this paper, we introduce *subset flows*, a class of flows that can tractably transform finite volumes and thus allow *exact* computation of likelihoods for discrete data. Based on subset flows, we identify ordinal discrete autoregressive models, including WaveNets, PixelCNNs and Transformers, as single-layer flows. We use the flow formulation to compare models trained and evaluated with either the exact likelihood or its dequantization lower bound. Finally, we study multilayer flows composed of PixelCNNs and non-autoregressive coupling layers and demonstrate state-of-the-art results on CIFAR-10 for flow models trained with dequantization.

## 1 Introduction

Learning generative models of high-dimensional data poses a significant challenge. The model will have to capture not only the marginal distributions of each of the variables, but also the potentially combinatorial number of interactions between them. Deep generative models provide tools for learning richly-structured, high-dimensional distributions, utilizing the vast amounts of unlabeled data available. Generative adversarial networks (GANs) (Goodfellow et al., 2014) are one class of deep generative models that have demonstrated an impressive ability to generate plausible-looking images. However, GANs typically lack support over the full data distribution and provide no quantitative measure of performance.

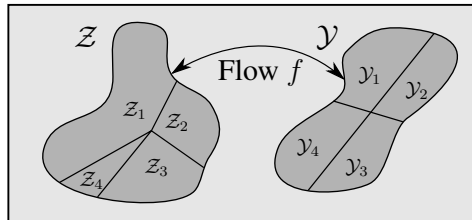

Figure 1: Subset flows $f : \mathcal{Y} \to \mathcal{Z}$ allow not only to transform points $\boldsymbol{z} = f(\boldsymbol{y})$, but also subsets $\mathcal{Z}_i = f(\mathcal{Y}_i)$, in one pass. As a result, these flows can be trained on ordinal discrete data without the need for dequantization.

*Likelihood-based deep generative models*, on the other hand, do provide this and can be classified as:

1. **Latent variable models** such as Deep Belief Networks (Hinton et al., 2006; Hinton, 2007), Deep Boltzmann Machines (Salakhutdinov and Hinton, 2009), Variational Autoencoders (VAEs) (Kingma and Welling, 2014; Rezende et al., 2014).

2. **Autoregressive models** such as Recurrent Neural Networks (RNNs), MADE (Germain et al., 2015), WaveNet (van den Oord et al., 2016a), PixelCNN (van den Oord et al., 2016c), PixelCNN++ (Salimans et al., 2017), Sparse Transformers (Child et al., 2019).

3. **Flow models** such as RealNVP (Dinh et al., 2017), Glow (Kingma and Dhariwal, 2018), MAF (Papamakarios et al., 2017), FFJORD (Grathwohl et al., 2019).

Data recorded from sensors are quantized before storage, resulting in ordinal data, i.e. discrete data with a natural ordering. Autoregressive models excel at modelling such data since they can directly model discrete distributions. Apart from discrete flows (Tran et al., 2019; Hoogeboom et al., 2019) – which are severely restricted in expressiveness – the vast majority of flow models are continuous and therefore require *dequantization* to be applied to discrete data. However, dequantization comes at the cost of lower bound estimates of the discrete likelihood (Theis et al., 2016; Ho et al., 2019).

In this paper: *1)* We introduce subset flows, a class of flows that allow tractable transformation of finite volumes and consequently may be trained directly on discrete data such as images, audio and video without the need for dequantization. *2)* Based on subset flows, we formulate existing autoregressive models for ordinal discrete data, such as PixelCNN (van den Oord et al., 2016c) and PixelCNN++ (Salimans et al., 2017), as single-layer autoregressive flows. *3)* Using the flow formulation of PixelCNNs, we quantify how dequantization used in training and evalutation impacts performance. *4)* We construct multilayer flows using compositions of PixelCNNs and coupling layers (Dinh et al., 2017). For CIFAR-10, we demonstrate state-of-the-art results for flow models trained with dequantization. The code used for experiments is publicly available at `https://github.com/didriknielsen/pixelcnn_flow`.

## 2 Background

**Normalizing flows** (Rezende and Mohamed, 2015) define a probability density $p(\boldsymbol{y})$ using an invertible transformation $f$ between $\boldsymbol{y}$ and a latent $\boldsymbol{z}$ with a base distribution $p(\boldsymbol{z})$, i.e.

$$\boldsymbol{y} = f^{-1}(\boldsymbol{z}) \quad \text{where} \quad \boldsymbol{z} \sim p(\boldsymbol{z}),$$

The density of $\boldsymbol{y}$ can be computed as

$$p(\boldsymbol{y}) = p(\boldsymbol{z}) \left| \det \frac{\partial \boldsymbol{z}}{\partial \boldsymbol{y}} \right| = p(f(\boldsymbol{y})) \left| \det \frac{\partial f(\boldsymbol{y})}{\partial \boldsymbol{y}} \right|.$$

The main challenge in designing flows is to develop transformations $f$ that are flexible, yet invertible and with cheap-to-compute Jacobian determinants. Luckily, more expressive flows can be obtained through a composition $f = f_K \circ ... \circ f_2 \circ f_1$ of simpler flow layers $f_1, f_2, ..., f_K$. The computation cost of the forward pass, the inverse pass and the Jacobian determinant for the composition will simply be the sum of costs for the components. While this compositional approach to building expressive densities make flow models attractive, they are not directly applicable to discrete data. Consequently, a method known as *dequantization* is typically employed.

**Uniform dequantization** refers to the process of converting discrete $\boldsymbol{x} \in \{0, 1, 2, ..., 255\}^D$ to a continuous $\boldsymbol{y} \in [0, 256]^D$ by simply adding uniform noise, i.e.

$$\boldsymbol{y} = \boldsymbol{x} + \boldsymbol{u} \quad \text{where} \quad \boldsymbol{u} \sim \prod_{d=1}^{D} \text{Unif}(u_d|0, 1).$$

This ensures that the values fill the continuous space $[0, 256]^D$ and consequently that continuous models will not collapse towards point masses at the discrete points during training. Uniform dequantization was proposed by Uria et al. (2013) with exactly this motivation. Theis et al. (2016) further showed that optimizing a continuous model on uniformly dequantized samples corresponds to maximizing a lower bound on a discrete log-likelihood.

**Variational dequantization** was introduced by Ho et al. (2019) as a generalization of uniform dequantization based on variational inference. Let $p(\boldsymbol{y})$ be some flexible continuous model and assume an observation model of the form $P(\boldsymbol{x}|\boldsymbol{y}) = \mathbb{I}(\boldsymbol{y} \in \mathcal{B}(\boldsymbol{x}))$, where $\mathcal{B}(\boldsymbol{x})$ is the region in $\mathcal{Y}$ associated with $\boldsymbol{x}$, e.g. a hypercube with one corner in $\boldsymbol{x}$, i.e. $\{\boldsymbol{x} + \boldsymbol{u}|\boldsymbol{u} \in [0, 1)^D\}$.

As shown by Ho et al. (2019), using a dequantization distribution $q(\boldsymbol{y}|\boldsymbol{x})$, one can develop a lower bound on the discrete log-likelihood $\log P(\boldsymbol{x})$ using Jensen's inequality,

$$\log P(\boldsymbol{x}) = \log \int P(\boldsymbol{x}|\boldsymbol{y})p(\boldsymbol{y})d\boldsymbol{y} = \log \int_{\boldsymbol{y} \in \mathcal{B}(\boldsymbol{x})} p(\boldsymbol{y})d\boldsymbol{y}$$

$$= \log \int_{\boldsymbol{y} \in \mathcal{B}(\boldsymbol{x})} q(\boldsymbol{y}|\boldsymbol{x}) \frac{p(\boldsymbol{y})}{q(\boldsymbol{y}|\boldsymbol{x})} d\boldsymbol{y} \geq \int_{\boldsymbol{y} \in \mathcal{B}(\boldsymbol{x})} q(\boldsymbol{y}|\boldsymbol{x}) \log \frac{p(\boldsymbol{y})}{q(\boldsymbol{y}|\boldsymbol{x})} d\boldsymbol{y}.$$

This corresponds exactly to the *evidence lower bound* (ELBO) used in variational inference, where the dequantization distribution $q(\boldsymbol{y}|\boldsymbol{x})$ coincides with the usual variational posterior approximation.

Note that for uniform dequantization, $q(\boldsymbol{y}|\boldsymbol{x}) = \prod_{d=1}^{D} \mathrm{Unif}(y_d|x_d, x_d + 1)$, the bound simplifies to $\log P(\boldsymbol{x}) \geq \mathbb{E}_{q(\boldsymbol{y}|\boldsymbol{x})}[\log p(\boldsymbol{y})]$ since $q(\boldsymbol{y}|\boldsymbol{x}) = 1$ over the entire integration region $\mathcal{B}(\boldsymbol{x})$. Training with this lower bound corresponds to the common procedure for training flows on discrete data, i.e. fit the continuous density $p(\boldsymbol{y})$ to uniformly dequantized samples $\boldsymbol{y}$. Ho et al. (2019) proposed to use a more flexible flow-based dequantization distribution $q(\boldsymbol{y}|\boldsymbol{x})$ in order to tighten the bound. The bound can further be tightened by using the importance weighted bound (IWBO) of Burda et al. (2016). In Sec. 3, we identify a class of flows which allow direct computation of $\log P(\boldsymbol{x})$ instead of a lower bound.

## 3 Closing the Dequantization Gap

In this section, we define the *dequantization gap*, the difference between the discrete log-likelihood and its variational lower bound due to dequantization. Next, we introduce *subset flows*, a class of flows for which dequantization is not needed, allowing us to directly optimize the discrete likelihood.

### 3.1 The Dequantization Gap

Flow models such as RealNVP (Dinh et al., 2017) and Glow (Kingma and Dhariwal, 2018) have achieved remarkable performance for image data while still allowing efficient sampling with impressive sample quality. However, in terms of log-likelihood, they still lag behind autoregressive models (Ho et al., 2019; Ma et al., 2019). While some of the performance gap might be the result of less expressive models, much of the gap seems to stem from a loose variational bound, as demonstrated by Ho et al. (2019) and Ma et al. (2019). We term the difference between the discrete log-likelihood and its lower bound the *dequantization gap*:

$$\mathrm{Deq.\ Gap} := \log P(\boldsymbol{x}) - \mathbb{E}_{q(\boldsymbol{y}|\boldsymbol{x})}\left[\log \frac{p(\boldsymbol{y})}{q(\boldsymbol{y}|\boldsymbol{x})}\right] = \mathbb{D}_{KL}\left[q(\boldsymbol{y}|\boldsymbol{x})\|p(\boldsymbol{y}|\boldsymbol{x})\right].$$

In the next subsection, we will introduce *subset flows* which allow the discrete likelihood to be computed in closed form. This completely closes the dequantization gap and allows us to recover existing autoregressive models as flow models.

### 3.2 Subset Flows

Dequantization facilitates computation of a lower bound of the discrete log-likelihood $\log P(\boldsymbol{x})$. However, using conservation of probability measure, we may compute the exact likelihood as

$$P(\boldsymbol{x}) = \int P(\boldsymbol{x}|\boldsymbol{y})p(\boldsymbol{y})d\boldsymbol{y} = \int_{\boldsymbol{y}\in\mathcal{B}(\boldsymbol{x})} p(\boldsymbol{y})d\boldsymbol{y} = \int_{\boldsymbol{z}\in f(\mathcal{B}(\boldsymbol{x}))} p(\boldsymbol{z})d\boldsymbol{z},$$

where $f(\mathcal{B}(\boldsymbol{x}))$ is the image of $f$ applied to $\mathcal{B}(\boldsymbol{x})$, i.e. $f(\mathcal{B}(\boldsymbol{x})) = \{f(\boldsymbol{y})|\boldsymbol{y} \in \mathcal{B}(\boldsymbol{x})\}$. Assuming a standard uniform base distribution, $p(\boldsymbol{z}) = \prod_{d=1}^{D} \mathrm{Unif}(z_d|0, 1)$, this formula takes the simple form

$$P(\boldsymbol{x}) = \int_{\boldsymbol{z}\in f(\mathcal{B}(\boldsymbol{x}))} d\boldsymbol{z} = \mathrm{Volume}(f(\mathcal{B}(\boldsymbol{x}))). \tag{1}$$

Interestingly, in order to compute $\log P(\boldsymbol{x})$, we do not need to keep track of infinitesimal volume changes with a Jacobian determinant. Instead, we have to keep track of the finite volume changes of the set $\mathcal{B}(\boldsymbol{x})$. While Eq. 1 applies to any flow $f$ in principle, the computation is generally intractable. We define *subset flows* as the class of flows $f : \mathcal{Y} \to \mathcal{Z}$ which have the additional property that they can tractably transform subsets of the input space, $\mathcal{Y}_s \subset \mathcal{Y}$, to subsets in the latent space, $\mathcal{Z}_s \subset \mathcal{Z}$, This is illustrated in Figure 1. By keeping track of how the finite volume $\mathcal{B}(\boldsymbol{x})$ is transformed to a finite volume $f(\mathcal{B}(\boldsymbol{x}))$ in the latent space, subset flows facilitate exact computation of the discrete likelihood in Eq. 1.

**Subset Flows in 1D.** In 1D, the computation in Eq. 1 with uniform $p(z)$ is particularly simple:

$$P(x) = \int P(x|y)p(y)dy = \int_{y=x}^{x+1} p(y)dy = \int_{z=f(x)}^{f(x+1)} p(z)dz = f(x + 1) - f(x),$$

where $f$ then must correspond to the *cumulative distribution function* (CDF) of $p(y)$.

**Autoregressive Subset Flows.** Subset flows present a different set of challenges compared to regular flows. In order to compute the discrete likelihood, we need not worry about computation of Jacobian determinants. Instead, we need flows where we can keep track of a finite volume. One straightforward approach to do this in higher dimensions is to work solely with hyperrectangles. Hyperrectangles have the benefit that they can easily be represented using two extreme points of the hyperrectangle. Furthermore, we can efficiently compute the volume of a hyperrectangle.

In order to work entirely with hyperrectangles, we need: 1) to partition the continuous space $\mathcal{Y}$ into hyperrectangles $\mathcal{B}(\boldsymbol{x})$ and 2) a flow $f$ such that the regions $f(\mathcal{B}(\boldsymbol{x}))$ resulting from $f$ remain hyperrectangles. For the first point with e.g. $\mathcal{Y} = [0, 256]^D$, we can define $\mathcal{B}(\boldsymbol{x}) = \{\boldsymbol{x} + \boldsymbol{u} | \boldsymbol{u} \in [0, 1)^D\}$, resulting in disjoint hypercubes for each of the discrete values. The second point can be achieved by using an autoregressive flow with what we denote *bin conditioning*.

**Bin conditioning** is achieved by conditioning on *the bin to which a value belongs rather than its exact value*. For the transformation of dimension $d$, this is achieved by

$$z_d^{(\text{lower})} = f\left(y_d^{(\text{lower})} | \boldsymbol{\lambda}_d\left(\boldsymbol{y}_{1:d-1}^{(\text{lower})}\right)\right),$$
$$z_d^{(\text{upper})} = f\left(y_d^{(\text{upper})} | \boldsymbol{\lambda}_d\left(\boldsymbol{y}_{1:d-1}^{(\text{lower})}\right)\right),$$

where $[y_d^{(\text{lower})}, y_d^{(\text{upper})}]$ are the boundaries of the input hyperrectangle and $[z_d^{(\text{lower})}, z_d^{(\text{upper})}]$ the output hyperrectangle. Importantly, the parameters $\boldsymbol{\lambda}_d$ are conditioned on the lower corner of the bin, $\boldsymbol{y}_{1:d-1}^{(\text{lower})}$, rather than the exact value $\boldsymbol{y}_{1:d-1}$, thus resulting in the same parameters $\boldsymbol{\lambda}_d$ regardless of the exact value of $\boldsymbol{y}_{1:d-1}$ within the bin. This is an instance of bin conditioning and ensures that the output region $f(\mathcal{B}(\boldsymbol{x}))$ will remain a hyperrectangle. Fig. 2 illustrates the effect of bin conditioning in a 2-dimensional binary problem. Note that conditioning on the upper corner or on both corners also constitute valid choices.

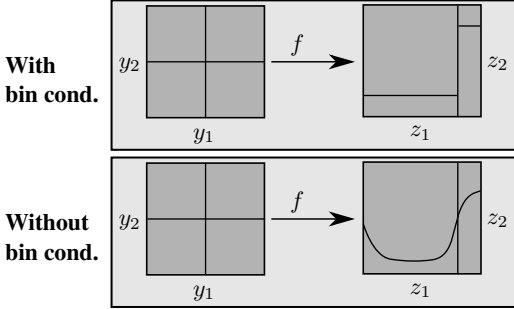

Figure 2: The effect of bin conditioning for a 2-dimensional binary problem. For the transformation with bin conditioning, the transformed rectangles remain rectangles.

# 4   PixelCNN as a Single-Layer Flow

In this section, we will show that several existing discrete autoregressive models, including WaveNet, PixelCNN and PixelCNN++, can be obtained as single-layer autoregressive flows, giving them a notion of a latent space and enabling their use as layers in a multi-layer flow.

**Autoregressive models** excel at modeling discrete data $\boldsymbol{x} \in \{0, 1, ..., 255\}^D$ such as images, audio and video since they can directly model discrete distributions. Numerous models of this form have been proposed in recent years (van den Oord et al., 2016c,b,a; Kalchbrenner et al., 2017; Salimans et al., 2017; Parmar et al., 2018; Chen et al., 2018; Menick and Kalchbrenner, 2019; Child et al., 2019). These models rely on autoregressive neural networks constructed using masked convolutions and/or masked self-attention layers and have constituted the state-of-the-art in terms of log-likelihood.

**PixelCNN** and related models (van den Oord et al., 2016c,b,a; Kalchbrenner et al., 2017; Menick and Kalchbrenner, 2019; Child et al., 2019) take a simple approach to modelling ordinal discrete data: they use autoregressive networks to parameterize Categorical distributions, i.e. $P(\boldsymbol{x}) = \prod_{d=1}^{D} \text{Cat}\left(x_d | \boldsymbol{x}_{1:d-1}\right)$. The Categorical distribution with $K$ categories may be obtained using subset flows as follows: Define a uniform base distribution, let $\mathcal{Y} = [0, K)$ and specify a piecewise linear CDF $f(y)$ (Müller et al., 2019),

$$f(y) = (y - (k-1))\pi_k + \sum_{l=1}^{k-1} \pi_l, \qquad \text{for } k-1 \leq y < k$$

where $\pi_1, ..., \pi_K \geq 0$, $\sum_{k=1}^{K} \pi_k = 1$. This yields a piecewise constant density $p(y)$, which upon quantization yields the Categorical distribution (see Fig. 3a). Using Eq. 1, we find the discrete

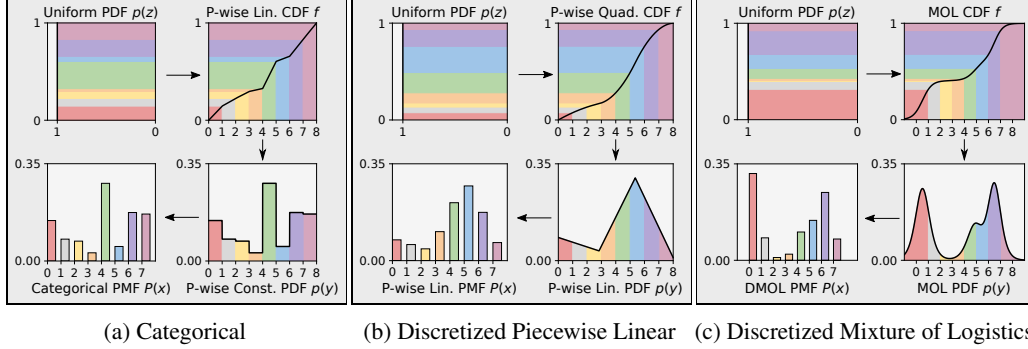

(a) Categorical  (b) Discretized Piecewise Linear  (c) Discretized Mixture of Logistics

Figure 3: Categorical, Discretized Piecewise Linear and Discretized Mixture of Logistics distributions as 1D subset flows. The *arrows* indicate the direction for **generating samples**: *1)* sample uniform noise $z$, *2)* pass $z$ through the inverse flow/CDF $f^{-1}$ to obtain a continuous sample $y$, *3)* quantize $y$ to obtain a discrete sample $x$. For subset flows, we can tractably invert this process to compute likelihoods. The *colors* illustrate the flow of mass when **computing the likelihood**: *1)* determine the region $\mathcal{B}(x)$ associated with observation $x$, *2)* pass the region through the flow (in 1D, pass the two extremes of the region through), *3)* compute the volume of the latent region. Note that while subset flows are straightforward in 1D, some care must be taken to extend them to higher dimensions.

likelihood to be $P(x) = f(x+1) - f(x) = \pi_k$. See App. A for more details. PixelCNN may thus be obtained as an autoregressive flow by using 1) a uniform base distribution, 2) bin conditioning, and 3) piecewise linear transformations, also known as linear splines, as the elementwise transformations. The result is an autoregressive subset flow which corresponds *exactly* to the original PixelCNN model.

**Higher order splines** such as quadratic, cubic or rational-quadratic (Müller et al., 2019; Durkan et al., 2019) can be used as replacement of the linear, yielding novel models. The distribution obtained from quadratic splines is illustrated in Fig. 3b (see App. A for more details). In our experiments, we show that quadratic splines tend to improve performance over linear splines.

**PixelCNN++** and related models (Salimans et al., 2017; Parmar et al., 2018; Chen et al., 2018) make use the *Discretized Mixture of Logistics* (DMOL) (Salimans et al., 2017) distribution, $P(\boldsymbol{x}) = \prod_{d=1}^{D} \text{DMOL}(x_d | \boldsymbol{x}_{1:d-1})$, The DMOL distribution can be obtained using subset flows as follows: Define a uniform base distribution and let $f$ be the CDF of a mixture of logistics distribution, i.e.

$$ f(y) = \sum_{m=1}^{M} \pi_m \sigma \left( \frac{y - 0.5 - \mu_m}{s_m} \right). $$

With bin boundaries defined at $y \in \{-\infty, 1, 2, ..., 255, \infty\}$, the discrete likelihood is

$$ P(x) = f(y^{(\text{upper})}) - f(y^{(\text{lower})}) = \begin{cases} \sum_{m=1}^{M} \pi_m \left[ \sigma \left( \frac{0.5 - \mu_m}{s_m} \right) \right], x = 0 \\ \sum_{m=1}^{M} \pi_m \left[ \sigma \left( \frac{x + 0.5 - \mu_m}{s_m} \right) - \sigma \left( \frac{x - 0.5 - \mu_m}{s_m} \right) \right], x = 1, ..., 254 \\ \sum_{m=1}^{M} \pi_m \left[ 1 - \sigma \left( \frac{255 - 0.5 - \mu_m}{s_m} \right) \right], x = 255 \end{cases} $$

corresponding exactly to the DMOL as defined in (Salimans et al., 2017) (illustrated in Fig. 3c).

In practice, PixelCNN++ makes use of a multivariate version of the DMOL distribution. For an image with $D = CS$ dimensions, $C$ channels and $S$ spatial locations, the $C$-dimensional distribution for each of the $S$ spatial locations are modelled using a multivariate DMOL distribution. This multivariate DMOL distribution may itself be expressed as an autoregressive flow. See App. B for more details. PixelCNN++ can thus be viewed as a *nested autoregressive flow* where the network is autoregressive over the spatial dimensions and outputs parameters for the autoregressive flows along the channels.

**Beyond single-layer flows.** By replacing the uniform base distribution by more flow layers, a more expressive distribution may be obtained. However, this will typically make the exact likelihood computation intractable, thus requiring dequantization. One exception is multi-layer autoregressive subset flows – where the autoregressive order is the same for all layers – which we consider in App. F. In our experiments, we show that compositions of PixelCNNs in flows yield powerful models.

## 5  Related Work

This work is related to several lines of work. First of all, this work builds on work formulating autoregressive models as flows (Kingma et al., 2016; Papamakarios et al., 2017; Huang et al., 2018; Oliva et al., 2018; Jaini et al., 2019). However, these only apply to continuous distributions and therefore do not include ordinal discrete autoregressive models such as the PixelCNN family of models (van den Oord et al., 2016c,b,a; Kalchbrenner et al., 2017; Salimans et al., 2017; Parmar et al., 2018; Chen et al., 2018; Menick and Kalchbrenner, 2019; Child et al., 2019).

Second, this work builds on the variational view of dequantization (Theis et al., 2016; Ho et al., 2019; Hoogeboom et al., 2020). Uria et al. (2013) introduced uniform dequantization, Theis et al. (2016) showed that this leads to a lower bound on the discrete likelihood and Ho et al. (2019) further proposed to use a more flexible dequantization distribution in order to tighten the dequantization gap. We expand on this by showing that for subset flows, we can perform exact inference and thus close the dequantization gap completely.

Finally, one may model discrete data with flows that are discrete. Hoogeboom et al. (2019) present discrete flows for ordinal integer data, while Tran et al. (2019) present discrete flows for nominal categorical data. Both of these works make use of the straight-through estimator (Bengio et al., 2013) to backpropagate through the rounding operations, resulting in a gradient bias. Unlike these works, we make use of continuous flows, but apply them to discrete data. Consequently, we can compute exact gradients and therefore avoid the performance impacts arising from biased gradients.

## 6  Experiments

### 6.1  The Latent Space of PixelCNNs

PixelCNN, PixelCNN++ and related models are typically viewed as purely discrete autoregressive models which have no latent space associated with them. Our interpretation of these models as single-layer flows opens up for exploration of these existing models latent space. To illustrate this possibility, we trained Pixel-CNN (van den Oord et al., 2016c) and Pixel-CNN++ (Salimans et al., 2017) as flow models on CIFAR-10, with results exactly matching the reported numbers of 3.14 and 2.92 bits/dim.

Some examples of interpolations between CIFAR-10 test images are shown in Figure 4. These were obtained by interpolating along a path of *equally-probable* samples under the base distribution. See App. E for more details.

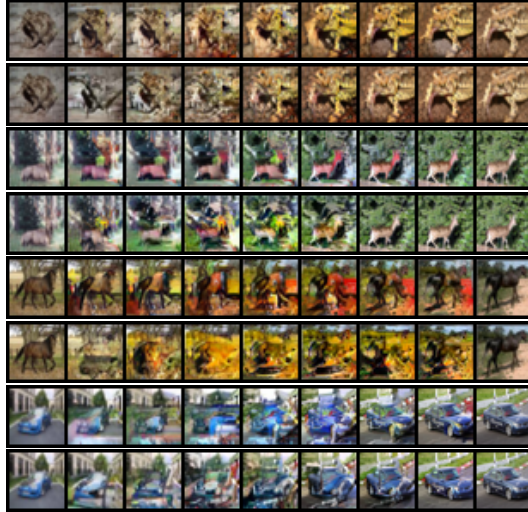

Figure 4: Latent space interpolations between pairs of CIFAR-10 test set images using PixelCNN (odd rows) and PixelCNN++ (even rows). These models are known for capturing local correlations well, but typically struggle with long-range dependencies. This is reflected in several of the interpolated images, which tend to lack global coherence.

### 6.2  The Effect of the Dequantization Gap

Our work shows that the PixelCNN family of models can be formulated as flow models where the dequantization gap between the true likelihood and the variational lower bound is completely closed. This suggests that they may be trained using either 1) uniform dequantization, 2) variational dequantization or 3) the exact likelihood. The resulting models should have decreasing dequantization gaps in the listed order. Surveying results from the literature (collected in Table 4 in App. C), we observe significant improvements between the categories with the best results for e.g. CIFAR-10 at 3.28, 3.08 and 2.80, suggesting that the dequantization gap has a significant impact on results.

Table 1: The effect of the dequantization gap. We compare three models, PixelCNN, PixelCNN (Quad.) and PixelCNN++. For each model, we trained three versions, one using the exact likelihood and two using the ELBO with uniform dequantization, both with and without bin conditioning. The models trained using the ELBO are evaluated using 1) the ELBO, 2) the IWBO (importance weighted bound) (Burda et al., 2016), and 3) the exact likelihood. See Sec. 6.2 for an explanation.

| Bin Cond. | Training | Eval. | PixelCNN | PixelCNN (Q) | PixelCNN++ |
|-----------|----------|-------|----------|--------------|------------|
| No | ELBO | ELBO | 3.248 | 3.251 | 3.112 |
|  |  | IWBO(10) | 3.235 | 3.237 | 3.095 |
|  |  | IWBO(100) | 3.227 | 3.228 | 3.086 |
|  |  | IWBO(1000) | 3.221 | 3.223 | 3.079 |
| Yes | ELBO | ELBO | **3.141** | 3.142 | 2.993 |
|  |  | IWBO(10) | **3.141** | 3.134 | 2.983 |
|  |  | IWBO(100) | **3.141** | 3.129 | 2.978 |
|  |  | IWBO(1000) | **3.141** | 3.126 | 2.974 |
|  |  | Exact | **3.141** | 3.104 | 2.944 |
| Yes | Exact | Exact | **3.141** | **3.090** | **2.924** |

To further test this hypothesis, we make use of our flow interpretation of existing autoregressive models. We train three flow models on CIFAR-10: 1) PixelCNN, 2) PixelCNN with quadratic splines (Quad.) and 3) PixelCNN++ using three different setups:

1. *Exact likelihood:* We train models exploiting the fact that for subset flows we can compute exact likelihoods.

2. *Dequantization w/ bin cond.:* In this case, we train the exact same models as before, but we replace the exact likelihood objective with the ELBO. With this setup, we can investigate:
   - The gap from the ELBO to the exact likelihood: $\log P(\boldsymbol{x}|\boldsymbol{\theta}_{\mathrm{ELBO}}) - \mathcal{L}(\boldsymbol{\theta}_{\mathrm{ELBO}})$.
   - How much closer the IWBO gets us in practice: $\log P(\boldsymbol{x}|\boldsymbol{\theta}_{\mathrm{ELBO}}) - \mathcal{L}_k(\boldsymbol{\theta}_{\mathrm{ELBO}})$.
   - The negative impact of training with the ELBO: $\log P(\boldsymbol{x}|\boldsymbol{\theta}_{\mathrm{Exact}}) - \log P(\boldsymbol{x}|\boldsymbol{\theta}_{\mathrm{ELBO}})$.

   Here, $\boldsymbol{\theta}$ denotes the model parameters, $\mathcal{L}(\boldsymbol{\theta})$ denotes the ELBO and $\mathcal{L}_k(\boldsymbol{\theta})$ the IWBO with $k$ importance samples for parameters $\boldsymbol{\theta}$. Furthermore, $\boldsymbol{\theta}_{\mathrm{ELBO}} = \arg\max_{\boldsymbol{\theta}} \mathcal{L}(\boldsymbol{\theta})$ and $\boldsymbol{\theta}_{\mathrm{Exact}} = \arg\max_{\boldsymbol{\theta}} \log P(\boldsymbol{x}|\boldsymbol{\theta})$.

3. *Dequantization w/o bin cond.:* We change the flows to not use bin conditioning. As a result, the latent regions will no longer be hyperrectangles and we therefore cannot compute exact likelihoods for these models. Note that this closely corresponds to how most flow models such as RealNVP and Glow are trained.

The results are given in Table 1. Some things to note from these results are:

- The exact models match the reported numbers in van den Oord et al. (2016c) and Salimans et al. (2017) at 3.14 and 2.92 bits/dim.

- Training with the ELBO negatively impacts performance, even when evaluating using the exact likelihood. Gaps of 0.014 and 0.020 bits/dim are found for PixelCNN (Quad.) and PixelCNN++.

- For models with bin conditioning trained with the ELBO, we can here compute the exact dequantization gap. For PixelCNN (Quad.) and PixelCNN++, this gap is found to be 0.038 and 0.049 bits/dim.

- The IWBO improves the estimate of $\log P(\boldsymbol{x})$ with an increasing number of importance samples. However, even for 1000 samples, less than half the gap has been closed, with 0.022 and 0.030 bits/dim remaining.

- For PixelCNN with bin conditioning, training with the ELBO does not impact performance. Here, the exact $p(\boldsymbol{y}|\boldsymbol{x})$ is uniform and therefore exactly matches the uniform dequantization distribution $q(\boldsymbol{y}|\boldsymbol{x})$, resulting in a dequantization gap of $\mathbb{D}_{KL}\left[q(\boldsymbol{y}|\boldsymbol{x})\|p(\boldsymbol{y}|\boldsymbol{x})\right] = 0$.

- Models trained without bin conditioning show significantly worse performance with gaps of 0.107, 0.161 and 0.188 to the original exact models. This shows that the usual approach for training flows using uniform dequantization and no bin conditioning leads to significantly worse performance.

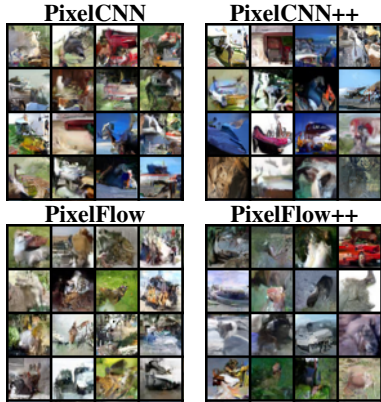

**PixelCNN**      **PixelCNN++**

**PixelFlow**      **PixelFlow++**

Figure 5: Unconditional samples.

| Model | AR | $\leq$Bits/dim |
|---|---|---|
| RealNVP (Dinh et al., 2017) | No | 3.49 |
| Glow (Kingma and Dhariwal, 2018) | No | 3.35 |
| Flow++ (Ho et al., 2019) | No | 3.08 |
| MintNet (Song et al., 2019) | Yes | 3.32 |
| MaCow (Cao et al., 2019) | Yes | 3.16 |
| PixelFlow (ours) | Yes | 3.04 |
| PixelFlow++ (ours) | Yes | **2.92** |

Table 3: Flow models trained with dequantization on CIFAR-10. PixelFlow(++) correspond to a composition of PixelCNN(++) and a Glow-like coupling flow.

### 6.3 PixelCNN in Flows

**PixelFlow.** We now demonstrate that compositions of PixelCNNs in flows can yield expressive models. We first construct 2 models which we term *PixelFlow* and *PixelFlow++*. PixelFlow++ is a composition of a PixelCNN++ and a multi-scale flow architecture (Dinh et al., 2017) using 2 scales with 8 steps/scale. Each step is a composition of a coupling layer (Dinh et al., 2017) and an invertible $1\times1$ convolution (Kingma and Dhariwal, 2018). Each coupling layer is parameterized by a DenseNet (Huang et al., 2017). PixelFlow uses the exact same setup as PixelFlow++, except it uses a quadratic spline version of PixelCNN instead of PixelCNN++. Both models make use of bin conditioning.

We train PixelFlow and PixelFlow++ using variational dequantization (Ho et al., 2019) and compare to other autoregressive and coupling flows trained with dequantization. The results are shown in Table 3. *PixelFlow++ obtains state-of-the-art results for flow models trained with dequantization on CIFAR-10.* Samples from PixelFlow and PixelFlow++ are shown in Fig. 5. More samples can be found in App. G.

**Stacks of PixelCNNs.** Next, we perform a series of experiments where we stack PixelCNNs in multi-layer flows with and without 90° rotations in-between. Table 2 shows results for stacks of 4 quadratic spline PixelCNNs. Note that when no rotation is used, the autoregressive order is the same for all the PixelCNNs. Consequently, we may use bin conditioning in all layers, yielding a multi-layer subset flow, which allows exact likelihood computation. As expected, the models using rotation tend to perform better than those without. Interestingly, however, the exact model *without rotation* performs on par with the variational dequantiza-

Table 2: A flow of 4 quadratic spline PixelCNNs trained on CIFAR-10 with or without 90° rotation using 1) uniform dequantization, 2) variational dequantization and 3) the exact likelihood.

| Rotation | Uni. | Var. | Exact |
|---|---|---|---|
| No | 3.066 | 3.026 | **3.012** |
| Yes | 3.058 | **3.012** | - |

tion model *with rotation*, which suffers from a non-zero dequantization gap. We further investigate multi-layer subset flows, which have dequantization gaps of exactly zero, in App. F. Further details on all experiments can be found in App. D.

## 7 Conclusion

We presented subset flows, a class of flows which can tractably transform finite volumes, a property that allow their use for ordinal discrete data like images and audio without the need for dequantization. Based on subset flows, we could explicitly formulate existing autoregressive models such as PixelCNNs and WaveNets as single-layer autoregressive flows. Using this formulation of PixelCNNs, we were able to quantify exactly the performance impacts of training and evaluating flow models using dequantization. We further demonstrated that expressive flow models can be obtained using PixelCNNs as layers in multi-layer flows.

Potential directions for future work include designing novel forms of subset flows and developing novel state-of-the-art flow architectures using the formulation of the PixelCNN family as flows.

## Acknowledgements

We thank Emiel Hoogeboom, Priyank Jaini, Shihan Wang and Sindy Löwe for helpful feedback.

## Funding Disclosure

This research was supported by the NVIDIA Corporation with the donation of TITAN X GPUs.

## Broader Impact

This is foundational research in generative models/unsupervised learning with proposal for new flow models and interpretation of existing autoregressive models as flow models. These models can be applied to for example unsupervised learning on images and audio. Unsupervised learning has the potential to greatly reduce the need for labeled data and thus improve models in applications such as medical imaging where a lack of data can be a limitation. However, it may also potentially be used to improve deep fakes with potentially malicious applications.

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
