[Supplementary Material]

# A  Linear and Quadratic Splines as Flows

For self-containedness, we will here summarize linear and quadratic spline flows, i.e. piecewise linear and quadratic flows, as presented by Müller et al. (2019).

## A.1  Linear Spline Flows

Consider a univariate flow $f : \mathcal{Y} \to \mathcal{Z}$ where $\mathcal{Y} = [0, Q]$ and the latent space $\mathcal{Z} = [0, 1]$. The transformation $f$ is piece-wise linear between the points $\{(y_k, z_k)\}_{k=0}^{K}$, where

$$0 \equiv y_0 < y_1 < y_2 < ... < y_K \equiv Q$$
$$0 \equiv z_0 < z_1 < z_2 < ... < z_K \equiv 1.$$

For piece-wise linear flows, we fix $y_1, y_2, ..., y_{K-1}$ and parameterize $z_1, z_2, ..., z_{K-1}$ using probabilities $\pi_1, \pi_2, ..., \pi_{K-1}$,

$$z_k = \sum_{l=1}^{k} \pi_l,$$

where $\sum_{l=1}^{K} \pi_l = 1$.

The forward, inverse and Jacobian determinant computation of a linear spline $f$ can be written as

$$\textbf{Forward:} \quad z = f(y) = \sum_{k=1}^{K} \mathbb{I}(y_{k-1} \le y < y_k) \left[ z_{k-1} + \pi_k \frac{y - y_{k-1}}{y_k - y_{k-1}} \right]$$

$$\textbf{Inverse:} \quad y = f^{-1}(z) = \sum_{k=1}^{K} \mathbb{I}(z_{k-1} \le z < z_k) \left[ y_{k-1} + (y - y_{k-1}) \frac{z - z_{k-1}}{\pi_k} \right]$$

$$\textbf{Jac. det.:} \quad \left| \det \frac{df(y)}{dy} \right| = \prod_{k=1}^{K} \pi_k^{\mathbb{I}(y \in [y_{k-1}, y_k))}.$$

### A.1.1  Relation to the Categorical Distribution

To obtain the Categorical distribution, we 1) let $Q = K$, 2) fix $y_0 = 0, y_1 = 1, ..., y_K = K$ and 3) use a uniform base distribution,

$$z \sim \text{Unif}(z|0, 1)$$
$$y = f^{-1}(z|\boldsymbol{\pi}).$$

This yields a piecewise constant density,

$$p(y|\boldsymbol{\pi}) = \prod_{k=1}^{K} \pi_k^{\mathbb{I}(y \in [k-1, k))},$$

which upon quantization yields the Categorical distribution,

$$p(x|\boldsymbol{\pi}) = \text{Cat}(x|\boldsymbol{\pi}) = \prod_{k=1}^{K} \pi_k^{\mathbb{I}(x=k)}.$$

## A.2  Quadratic Spline Flows

Consider again a univariate flow $f : \mathcal{Y} \to \mathcal{Z}$ where $\mathcal{Y} = [0, Q]$ and the latent space $\mathcal{Z} = [0, 1]$. The transformation $f$ is piece-wise quadratic between the points $\{(y_k, z_k)\}_{k=0}^{K}$, where

$$0 \equiv y_0 < y_1 < y_2 < ... < y_K \equiv Q$$
$$0 \equiv z_0 < z_1 < z_2 < ... < z_K \equiv 1.$$

In the linear spline case, we fixed the locations $y_0, y_1, ..., y_K$. If this is not done, the objective becomes discontinuous and thus difficult to train with gradient-based optimizers. In the quadratic

case, however, we may let the bin locations $y_0, y_1, ..., y_K$ be free parameters to be learned (Müller et al., 2019).

The parameters of the flow are given by vectors $\hat{\boldsymbol{w}} \in \mathbb{R}^K$ and $\hat{\boldsymbol{v}} \in \mathbb{R}^{K+1}$. The bin widths are computed as

$$\boldsymbol{w} = Q \cdot \text{softmax}(\hat{\boldsymbol{w}}),$$

while the bin edges are computed as

$$\boldsymbol{v} = \frac{\exp(\hat{\boldsymbol{v}})}{\sum_{k=1}^{K} \frac{\exp(\hat{v}_k)+\exp(\hat{v}_{k+1})}{2} w_k}.$$

The bin locations are given by the sum of bin widths

$$y_k = \sum_{l=1}^{k} w_l.$$

For a quadratic spline, the density will be piece-wise linear. We can use this to compute the mass in bin $k$ between the lower extreme $y_{k-1}$ and some point $y^*$ can be computed as

$$\int_{y=y_{k-1}}^{y^*} \left[ v_{k-1} + \frac{y - y_{k-1}}{y_k - y_{k-1}}(v_k - v_{k-1}) \right] dy = \int_{t=0}^{\alpha} \left[ v_{k-1} + t(v_k - v_{k-1}) \right](y_k - y_{k-1})dt$$

$$= \left[ tv_{k-1} + \frac{1}{2}t^2(v_k - v_{k-1}) \right]_{t=0}^{\alpha} (y_k - y_{k-1})$$

$$= \left[ \alpha v_{k-1} + \frac{1}{2}\alpha^2(v_k - v_{k-1}) \right] w_k.$$

where $t \equiv \frac{y - y_{k-1}}{y_k - y_{k-1}}$ and $\alpha \equiv \frac{y^* - y_{k-1}}{y_k - y_{k-1}}$. The total mass in the bin can be found by setting $y^* = y_k$ or equivalently $\alpha = 1$ to obtain

$$\int_{y=y_{k-1}}^{y_k} \left[ v_{k-1} + \frac{y - y_{k-1}}{y_k - y_{k-1}}(v_k - v_{k-1}) \right] dy = \frac{v_{k-1} + v_k}{2} w_k$$

Using this, we find that

$$z_k = \sum_{l=1}^{k} \frac{v_{l-1} + v_l}{2} w_l$$

The forward, inverse and Jacobian determinant computation of the resulting quadratic spline flow $f$ can be written as

**Forward:** $\quad z = f(y) = \sum_{k=1}^{K} \mathbb{I}(y_{k-1} \le y < y_k) \left[ z_{k-1} + w_k \left( \alpha_k v_{k-1} + \frac{1}{2}\alpha_k^2(v_k - v_{k-1}) \right) \right]$

**Inverse:** $\quad y = f^{-1}(z) = \sum_{k=1}^{K} \mathbb{I}(z_{k-1} \le z < z_k) \left[ y_{k-1} + w_k \frac{\sqrt{v_{k-1}^2 + 2\frac{v_k - v_{k-1}}{w_k}(z - z_{k-1})} - v_{k-1}}{v_k - v_{k-1}} \right]$

**Jac. det.:** $\quad \left| \det \frac{df(y)}{dy} \right| = [v_{k-1} + \alpha_k(v_k - v_{k-1})]^{\mathbb{I}(y \in [y_{k-1}, y_k))}$,

where

$$\alpha_k = \frac{y - y_{k-1}}{y_k - y_{k-1}}.$$

# B  The Multivariate DMOL as a Flow

We will show that the *multivariate discretized mixture of logistics* (multivariate DMOL) distribution can be obtained as an autoregressive flow. First, we describe the distribution as it was presented in Salimans et al. (2017).

## B.1 The Multivariate DMOL Distribution

The discretized logistic distribution can be written as

$$P(x) = \text{DLogistic}(x|\mu, s) = \begin{cases} \sigma\left(\frac{0.5-\mu}{s}\right), x = 0. \\ \sigma\left(\frac{x+0.5-\mu}{s}\right) - \sigma\left(\frac{x-0.5-\mu}{s}\right), x = 1, ..., 254. \\ 1 - \sigma\left(\frac{255-0.5-\mu}{s}\right), x = 255. \end{cases}$$

The univariate DMOL distribution uses this as mixture components,

$$P(x) = \text{DMOL}(x|\boldsymbol{\pi}, \boldsymbol{\mu}, \boldsymbol{s}) = \sum_{m=1}^{M} \pi_m \text{DLogistic}(x|\mu_m, s_m).$$

The multivariate DMOL distribution, on the other hand, can be written as

$$P(\boldsymbol{x}) = \text{MultiDMOL}(\boldsymbol{x}|\boldsymbol{\pi}, \boldsymbol{\mu}, \boldsymbol{s}, \boldsymbol{r}) = \sum_{m=1}^{M} \pi_m \text{DLogistic}(x_3|\mu_{3,m}(x_1, x_2, \boldsymbol{r}_m), s_{3,m})$$
$$\text{DLogistic}(x_2|\mu_{2,m}(x_1, \boldsymbol{r}_m), s_{2,m})$$
$$\text{DLogistic}(x_1|\mu_{1,m}, s_{1,m}),$$

where

$$\mu_{1,m} = \mu_{1,m}$$
$$\mu_{2,m}(x_1, \boldsymbol{r}_m) = \mu_{2,m} + r_{1,m}x_1 \tag{2}$$
$$\mu_{3,m}(x_1, x_2, \boldsymbol{r}_m) = \mu_{3,m} + r_{2,m}x_1 + r_{3,m}x_2.$$

## B.2 Rewriting the Multivariate DMOL Distribution

We will now show how one can rewrite this distribution in an autoregressive form. Consider the usual 3-dimensional case and write the multivariate DMOL as

$$P(\boldsymbol{x}) = \sum_{m=1}^{M} \pi_m P_m(x_3|x_2, x_1) P_m(x_2|x_1) P_m(x_1).$$

We can rewrite this distribution as

$$P(\boldsymbol{x}) = \sum_{m=1}^{M} \pi_m P_m(x_3|x_2, x_1) P_m(x_2|x_1) P_m(x_1)$$
$$= \left[\sum_{m=1}^{M} \pi_{3,m} P_m(x_3|x_2, x_1)\right] \left[\sum_{m=1}^{M} \pi_{2,m} P_m(x_2|x_1)\right] \left[\sum_{m=1}^{M} \pi_{1,m} P_m(x_1)\right]$$

where

$$\pi_{1,m} = \pi_m$$
$$\pi_{2,m} = \frac{\pi_m P_m(x_1)}{\sum_{m'=1}^{M} \pi_{m'} P_{m'}(x_1)} \tag{3}$$
$$\pi_{3,m} = \frac{\pi_m P_m(x_2|x_1) P_m(x_1)}{\sum_{m'=1}^{M} \pi_{m'} P_{m'}(x_2|x_1) P_{m'}(x_1)}.$$

## B.3 The Multivariate DMOL Flow

To summarize, we can write the multivariate DMOL as an autoregressive distribution with univariate DMOL conditionals,

$$\text{MultiDMOL}(\boldsymbol{x}|\boldsymbol{\pi}, \boldsymbol{\mu}, \boldsymbol{s}, \boldsymbol{r}) = \text{DMOL}(x_3|\boldsymbol{\pi}(x_2, x_1), \boldsymbol{\mu}(x_2, x_1, \boldsymbol{r}), \boldsymbol{s})$$
$$\cdot \text{DMOL}(x_2|\boldsymbol{\pi}(x_1), \boldsymbol{\mu}(x_1, \boldsymbol{r}), \boldsymbol{s})$$
$$\cdot \text{DMOL}(x_1|\boldsymbol{\pi}, \boldsymbol{\mu}, \boldsymbol{s}),$$

where the means are given by Eq. 2 and the mixture weights by Eq. 3. Thus, we can obtain the multivariate DMOL flow as an autoregressive flow with the univariate DMOL flows from Sec. 3.2 as elementwise transformations.

## C  A Collection of Results from Previous Work

We here collect results from existing work in order to compare flow models trained with 1) uniform dequantization, 2) variational dequantization and 3) exact likelihood. The categories should have decreasing dequantization gaps in the listed order. The results are shown in Table 4. We observe significant improvements between the categories, suggesting that the dequantization gap has a significant impact on results.

Table 4: A collection of results from previous work (in bits/dim). We here divide the results into three categories, those trained with: 1) uniform dequantization, 2) variational dequantization and 3) exact likelihood. We observe that performance tends to drastically improve between the categories, suggesting the importance of the dequantization gap for flow models.

| Training | Model | CIFAR-10 | ImageNet32 | ImageNet64 |
|---|---|---|---|---|
| ELBO (U) | RealNVP (Dinh et al., 2017) | 3.49 | 4.28 | 3.98 |
| | Glow (Kingma and Dhariwal, 2018) | 3.35 | **4.09** | 3.81 |
| | MaCow (Ma et al., 2019) | **3.28** | - | **3.75** |
| | Flow++ (Ho et al., 2019) | 3.29 | - | - |
| ELBO (V) | MaCow (Ma et al., 2019) | 3.16 | - | **3.69** |
| | Flow++ (Ho et al., 2019) | **3.08** | 3.86 | 3.69 |
| Exact | PixelCNN (van den Oord et al., 2016c) | 3.14 | - | - |
| | Gated PixelCNN (van den Oord et al., 2016b) | 3.03 | 3.83 | 3.57 |
| | PixelCNN++ (Salimans et al., 2017) | 2.92 | - | - |
| | Image Transformer (Parmar et al., 2018) | 2.90 | **3.77** | - |
| | PixelSNAIL (Chen et al., 2018) | 2.85 | 3.80 | 3.52 |
| | SPN (Menick and Kalchbrenner, 2019) | - | 3.79 | 3.52 |
| | Sparse Transformer (Child et al., 2019) | **2.80** | - | **3.44** |

## D  Experimental Details

The code used for experiments is publicly available[1]. In our experiments, we used PixelCNN (van den Oord et al., 2016c) and PixelCNN++ (Salimans et al., 2017) models. For hyperparameters, we followed the original architectures as closely as possible. The PixelCNN architecture we used for the CIFAR-10 experiments is summarized in Table 5.

Table 5: The PixelCNN architecure used for the CIFAR-10 dataset.

| 3 x 32 x 32 RGB Image |
|---|
| Conv7x7(256) (Mask A) |
| **15 Residual Blocks, each using:** |
| ReLU - Conv1x1(256) (Mask B) |
| ReLU - Conv3x3(128) (Mask B) |
| ReLU - Conv1x1(256) (Mask B) |
| ReLU - Conv1x1(1024) (Mask B) |
| ReLU - Conv1x1(P) (Mask B) |

For the PixelCNN++ we used, like Salimans et al. (2017), 6 blocks with 5 ResNet layers. Between blocks 1 and 2 and between blocks 2 and 3, strided convolutions are used to downsample the feature maps. Between blocks 4 and 5 and between blocks 5 and 6, transposed strided convolutions are used to upsample the feature maps back to the original size. Shortcut connections are added from block 1 to 6, 2 to 5 and 3 to 4. For more details on the PixelCNN++ architecture see Salimans et al. (2017) and their publicly available code[2].

For models using linear splines we used 256 bins, corresponding to the quantization level. For models using quadratic splines, we used 16 bins as this was found to work well in early experiments. Note that for quadratic splines, the bin locations can also be learned (Müller et al., 2019). As a

consequence, less bins are typically required than for linear splines. Finally, for models using DMOL, 10 mixtures were used.

PixelFlow and PixelFlow++ use compositions of a PixelCNN(++) and a Glow-like coupling flow. The coupling flow is a multi-scale architecure (Dinh et al., 2017) using 2 scales and 8 steps/scale. Each step consists of an affine coupling layer (Dinh et al., 2017) and an invertible $1 \times 1$ convolution (Kingma and Dhariwal, 2018). The models were trained with variational dequantization (Ho et al., 2019) and an initial squeezing layer (Dinh et al., 2017) to increase the number of channels from 3 to 12. The coupling layers are parameterized by DenseNets (Huang et al., 2017).

All models were trained for the CIFAR-10 dataset were trained for 500 epochs with a batch size of 16. For all models, except PixelFlow and PixelFlow++, the Adam optimizer (Kingma and Ba, 2015) was used with an initial learning rate of $3 \cdot 10^{-4}$. For PixelFlow and PixelFlow++, the Adamax optimizer (Kingma and Ba, 2015) was used with an initial learning rate of $1 \cdot 10^{-3}$. For all models, the learning rate was decayed by $0.5$ at epochs $250, 300, 350, 400, 450$. All models were trained on a single GPU. Depending on the type of GPU used, training a stock PixelCNN takes about 30 hours and training a stock PixelCNN++ takes about 10 days.

# E    Interpolation Experiment Details

In Fig. 4, latent space interpolations in PixelCNN and PixelCNN++ models are shown. In order to obtain these interpolations, we first transform two real images $\boldsymbol{x}^{(0)}$ and $\boldsymbol{x}^{(1)}$ to the latent regions $f(\mathcal{B}(\boldsymbol{x}^{(0)}))$ and $f(\mathcal{B}(\boldsymbol{x}^{(1)}))$ and sample according to the uniform base distribution to obtain points $\boldsymbol{z}^{(0)}$ and $\boldsymbol{z}^{(1)}$ in the latent space. Linearly interpolating in this space does not yield uniform samples. Empirically, we found this to often give blurry, single-color interpolations. To get interpolated points that are valid samples from the uniform distribution, we first further transformed the latent images $\boldsymbol{z}^{(0)} \to \boldsymbol{h}^{(0)}$ and $\boldsymbol{z}^{(1)} \to \boldsymbol{h}^{(1)}$ using the inverse Gaussian CDF for each dimension. As this is an invertible transformation, we can equivalently consider the base distribution as the isotropic Gaussian. Subsequently, we interpolated according to

$$ \boldsymbol{h}^{(w)} = \frac{w\boldsymbol{h}^{(0)} + (1-w)\boldsymbol{h}^{(1)}}{\sqrt{w^2 + (1-w)^2}}, $$

for $0 \leq w \leq 1$. This yields a path of equally probable samples under the base distribution, i.e. $\boldsymbol{h}^{(w)} \sim \mathrm{N}(0,1)$ for $\boldsymbol{h}^{(0)}, \boldsymbol{h}^{(1)} \sim \mathrm{N}(0,1)$. Finally, the intermediate latent points $\boldsymbol{h}^{(w)}$ are transformed back to samples $\boldsymbol{x}^{(w)}$.

# F    Multilayer Subset Flows Experiments

In this section, we investigate multi-layer subset flows. Multi-layer subset flows can tractably transform finite volumes through *multiple* transformations. For *autoregressive subset flows*, this may be achieved by letting all the autoregressive flows share the same autoregressive order.

Denote the intermediate spaces as $\boldsymbol{z}_l, l = 0, 1, ..., L$ with $\boldsymbol{y} \equiv \boldsymbol{z}_0$ and $\boldsymbol{z} \equiv \boldsymbol{z}_L$. In a layer $l$, we further denote the boundaries of the hyperrectangle for dimension $d$ as $z_{d,l}^{(\mathrm{lower})}$ and $z_{d,l}^{(\mathrm{upper})}$. The procedure for computing the likelihood in a multilayer autoregressive subset flow is shown in Algo. 1, while the procedure for sampling is shown in Algo. 2.

Table 6: Multiple layers of subset flows can be used to obtain more expressive models. We here illustrate this by stacking vanilla PixelCNNs parameterizing either linear or quadratic splines in multiple layers for CIFAR-10. Reported numbers are in bits/dim and training performance in parentheses.

| Layers | Dropout | PixelCNN | PixelCNN (Quad.) |
|--------|---------|----------|------------------|
| 1 | - | 3.14 (3.11) | 3.09 (3.05) |
| 2 | - | 3.07 (2.98) | 3.05 (2.96) |
| 4 | - | 3.09 (2.89) | 3.09 (2.88) |
| | 0.2 | **3.02** (2.98) | **3.01** (2.98) |

---

**Algorithm 1** Multilayer Subset Flow Likelihood

Observe discrete $\boldsymbol{x}$.
Define $\boldsymbol{z}_0^{(\text{lower})} = \boldsymbol{x}, \boldsymbol{z}_0^{(\text{upper})} = \boldsymbol{x} + 1$.
**for** $l = 1$ **to** $L$ **do**
  Compute $\boldsymbol{\lambda}_l \leftarrow \text{Net}\left(\boldsymbol{z}_{l-1}^{(\text{lower})}\right)$.
  Transform $\boldsymbol{z}_l^{(\text{lower})} = f_l(\boldsymbol{z}_{l-1}^{(\text{lower})}|\boldsymbol{\lambda}_l)$.
  Transform $\boldsymbol{z}_l^{(\text{upper})} = f_l(\boldsymbol{z}_{l-1}^{(\text{upper})}|\boldsymbol{\lambda}_l)$.
**end for**
$\log P(\boldsymbol{x}) = \sum_{d=1}^{D} \log\left[z_{L,d}^{(\text{upper})} - z_{L,d}^{(\text{lower})}\right]$.

---

**Algorithm 2** Multilayer Subset Flow Sampling

Sample $\boldsymbol{z} \sim \prod_{d=1}^{D} \text{Unif}(z_d|0,1)$.
**for** $d = 1$ **to** $D$ **do**
  **for** $l = 1$ **to** $L$ **do**
    Compute $\boldsymbol{\lambda}_{l,d} \leftarrow \text{Net}\left(\boldsymbol{z}_{l-1,1:d-1}^{(\text{lower})}\right)$.
    Transform $\boldsymbol{z}_{l-1,d} = f_{l,d}^{-1}(z_{l,d}|\boldsymbol{\lambda}_{l,d})$.
  **end for**
  Define $y_d = z_{0,d}$.
  Quantize $y_d$ to obtain $x_d$.
**end for**

---

### F.1 Multilayer PixelCNN Subset Flows for CIFAR-10

We will here demonstrate that performance improves with an increasing number of PixelCNNs in a multi-layer subset flow, even when they all share the same autoregressive order. Due to its relatively lightweight nature compared to more recent autoregressive models, we use the original architecture of van den Oord et al. (2016c). We train 1, 2 and 4 layers of PixelCNN parameterizing either linear or quadratic splines on CIFAR-10. The results can be found in Table 6.

The single-layer linear spline case corresponds to the original PixelCNN model and the performance reported here matches van den Oord et al. (2016c) with 3.14 bits/dim. We observe that increasing the number of layers improve the fit over the single-layer version. In fact, the 4-layer version is flexible enough to overfit. By countering this using dropout with a rate of 0.2, the test set performance of the 4-layer version greatly improves, yielding the best performing model. In fact, by simply stacking 4 layers of the original PixelCNN, we outperform the improved model Gated PixelCNN (van den Oord et al., 2016b). In addition to multiple layers improving performance, we observe that quadratic splines improve performance over linear splines in all cases here.

### F.2 Multilayer PixelCNN++ Subset Flows for CIFAR-10

We here experiment with combining PixelCNN++ (Salimans et al., 2017) with other models in a multilayer subset flow. We use the original PixelCNN++ architecture and combine this with the original PixelCNN architecture (van den Oord et al., 2016c) parameterizing quadratic splines with 16 bins. We train this combination in both orders. The PixelCNN architecture here uses a dropout rate of 0.5, matching that of the PixelCNN++ architecture. The results are shown in Table 7.

We observe some slight improvements from combining PixelCNN++ with other models. Earlier experiments indicated that improving on PixelCNN++ by combining it with more layers of PixelCNN or PixelCNN++ was difficult since this often resulted in severe overfitting. We thus observe only mild improvements over the original model by combining it with other flows. We note that these are the best reported results for CIFAR-10 among models that only use convolutions and do not rely on self-attention. For further improvements, incorporating self-attention in the model would thus probably be of help.

Table 7: PixelCNN++ with combined with PixelCNN (Quad.) for the CIFAR-10 dataset.

| Model | Bits/dim |
|---|---|
| PixelCNN++ | 2.924 |
| PixelCNN++ & PixelCNN (Quad.) | 2.914 |
| PixelCNN (Quad.) & PixelCNN++ | **2.906** |

### F.3 Multilayer PixelCNN Subset Flows for ImageNet 32x32 and 64x64

In addition to training PixelCNN (van den Oord et al., 2016c) models in multiple flow layers for CIFAR-10, we here attempt the same for the 32x32 and 64x64 ImageNet datasets. For simplicity, all parameters were kept the same as in the case of CIFAR-10, except that 20 residual blocks were used for ImageNet64 (due to the need of a larger receptive field). The results for PixelCNN, $2\times$PixelCNN and $2\times$PixelCNN (Quad.) are shown in Table 8. We observe that performance improves with 2 layers instead of 1 and that swapping linear splines with quadratic splines further improves performance. Note that the performance of these models are slightly sub-par compared to other results by autoregressive models for these datasets. We attribute this to the fact that most other work use larger models, such as more filters in the convolutional layers, than the simple PixelCNN architectures we used here. Regardless, these results show that stacking more layers also helps for the 32x32 and 64x64 ImageNet datasets.

The ImageNet32 models were trained for 30 epochs with a batch size of 16, while the ImageNet64 models were trained for 20 epochs with a batch size of 8. The Adam optimizer (Kingma and Ba, 2015) was used. The learning rate was initially $3 \cdot 10^{-4}$ and decayed by 0.5 at epochs $15, 18, 21, 24, 27$ (for ImageNet32) and epochs $10, 12, 14, 16, 18$ (for ImageNet64).

Table 8: Multiple layers of vanilla PixelCNNs for the 32x32 and 64x64 ImageNet datasets.

| Model | ImageNet32 | ImageNet64 |
|---|---|---|
| PixelCNN | 3.96 | 3.67 |
| $2 \times$ PixelCNN | 3.91 | 3.63 |
| $2 \times$ PixelCNN (Quad.) | **3.90** | **3.61** |

## G Samples

See Fig. 6 for samples from PixelCNN, PixelFlow, PixelCNN++ and PixelFlow++.

(a) PixelCNN.

(b) PixelCNN++.

(c) PixelFlow.

(d) PixelFlow++.

Figure 6: Unconditional samples from PixelCNN-based flow models trained on the CIFAR-10 dataset. The PixelFlow(++) model composes PixelCNN(++) with a Glow-like coupling flow. The perceptual quality appears to improve in the multi-layer flow models compared to the single-layer base models.

## Footnotes

[1]https://github.com/didriknielsen/pixelcnn_flow

[2]https://github.com/openai/pixel-cnn