[Reviews · NeurIPS 2020]

Review 1

Summary and Contributions: The paper considers transformations of finite volumes of Euclidean space, and provides a perspective in which discrete autoregressive models (PixelCNN(++), WaveNet, the Transformer family) can be interpreted as single-layer flows. With this perspective, the paper then presents a comparison with other more typical flow methods which operate on continuous data, transform individual points rather than finite volumes, and are traditionally adapted to work with discrete data through dequantization.

Strengths: The perspective of discrete autoregressive models as single-layer flows places these models nicely relative to the large flow literature which has emerged in the past few years. It also presents a unified discussion of dequantization for ordinal data in this context, which until recently had been treated as somewhat of a heuristic to prevent degenerate solutions when fitting continuous models to discrete data. I would also argue the paper is particularly well-written in laying out this perspective. It neatly ties together various parts of the literature, and makes use of some nice figures (1, 2, 3) to aid understanding. There is also real value in the empirical evaluation of section 6.2 and Table 1. Thus far a direct examination of this so-called 'dequantization gap' has lacked in the literature, and it's an evaluation which I wished had been carried out. I am glad to see these comparisons included here.

Weaknesses: Ultimately, my main issue with the paper is that, despite proposing a new class of 'subset flows', there is a sense in which existing work has just been relabeled, with not much significantly new added. If we are to realise discrete autoregressive models, as exemplified by the PixelCNN family, as instances of subset flows, we might expect subset flows to do more than just offer a new perspective, namely suggest new developments which extend beyond what is already established in the literature. The equivalence between a categorical pdf and a linear spline transformation (the corresponding cdf) ties in with recent work on spline transformations in flows, but again there's really not a whole lot of value in considering the quadratic spline extension: the advantage of the DMOL parameterization over the categorical parameterization lies in the inductive bias that nearby values are assigned similar probabilities (compare piecewise constant pdf in bottom right of Fig 3 (a) vs (c)), which is helpful for ordinal data. The quadratic cdf with piecewise linear pdf (bottom right Fig 3 (b)) also has this inductive bias, but is restricted to be piecewise linear (strictly less general than DMOL). This ordering of flexibility is empirically confirmed by the results in Table 1, so I don't see why the quadratic spline is to be preferred over DMOL. Moreover, if discrete PixelCNNs are one-layer subset flows, then you'd expect a natural extension to multi-layer flows to be presented, but this isn't the case. Indeed, (178-179): 'By replacing the uniform base distribution by more flow layers, a more expressive distribution may be obtained. However, this will typically make the exact likelihood computation intractable, thus requiring dequantization'. This happens because reordering the variables between layers make it very difficult (maybe impossible?) to track the transformed volumes, and without re-ordering, multiple layers are equivalent to a single layer. Solving this problem to make subset flows a meaningful extension to the existing literature would be significant contribution, rather than leaving it to future work (304-305).

Correctness: The paper is technically sound, and the empirical evaluation is generally fine. I'm not entirely sure what the interpolations in section 6.1 are meant to demonstrate. I'm also not sure why trying to achieve SOTA results using dequantization in section 6.3 is all that relevant -- isn't the main thrust of the paper to 'close the dequantization gap'? Creating a flow with both autoregressive and coupling transforms seems a little odd. Why would this architecture be preferred over e.g. a large sparse transformer, or even a large PixelCNN?

Clarity: In laying out a cohesive narrative between existing aspects of the literature and the associated concepts (flows, dequantization, discrete autoregressive models), I feel the paper is particularly well written.

Relation to Prior Work: As discussed already, the paper explicitly distinguishes 'subset flows' from prior work, and demonstrates how this prior work can be seen as a particular instance of 'subset flows', but the lack of further development of 'subset flows' makes it somewhat difficult to pin down the real contribution here.

Reproducibility: Yes

Additional Feedback: -------------------------------------------- POST-REBUTTAL UPDATE -------------------------------------------- I'd like to thank the authors for their response. I think the value of this paper lies mainly in its empirical investigation of the effects of various dequantization schemes vs fully discrete models, and its well written exposition of the related literature. However, the proposed 'subset flows' don't expand the range of models we have available, since stacking these transforms without variable reordering between layers is equivalent to a single transform, and reordering means we lose the ability to track the volumes. Overall I would lean towards acceptance.


Review 2

Summary and Contributions: This paper analyses the role of dequantization (i.e. transforming discrete data to continuous data) in flow based models. This dequantization step typically means that models end up estimating a lower bound on the log likelihood of the true discrete data, as opposed to the exact log likelihood. To overcome this, the authors introduce a new type of flow layer that allows for dequantization without sacrificing exact log likelihoods. The authors do this by first assigning each discrete data point to a hypercube in the continuous space. For example, for images the data typically lies in {0, 1, …, 255}^D, and so each point in this space is assigned to a hypercube in the continuous [0, 256]^D space. The authors then define a flow layer that, instead of mapping points to points, maps each of these hypercubes to another hyperrectangle (i.e. a subset to another subset, hence the name subset flows). Crucially, the volume change from this operation is tractable and can be used to compute exact log likelihoods. The authors then show how this partitioning of the space and subsequent bin conditioning (the name given to the transformation of each hypercube) has similarities to autoregressive models like PixelCNNs. Indeed, the authors show that for certain choices of base distribution and transformations the model becomes exactly the same as a PixelCNN or a PixelCNN++ model. The authors use this connection to endow PixelCNNs with a type of latent space which can be used for e.g. image interpolation and so on. As PixelCNNs can then be interpreted as a layer of a flow, the authors also propose using a PixelCNN layer as a step in a flow model, effectively allowing for exact log likelihood with dequantization for flow based models. The authors show through experiments that their proposed layer provides a boost in performance over models with uniform or variational dequantization. They also provide thorough ablation studies showing the effects of different dequantization techniques which are interesting. Finally the authors show that combining their model with other flow layers can also boost performance in certain cases. Contributions: -The introduction of subset flows which allow for training flow models without requiring dequantization and therefore allowing for exact log likelihoods - Showing a connection between their proposed layer and autoregressive models like PixelCNNs, showing that these are effectively single layer flows - An experimental study of the effects of different types of dequantization on log likelihood scores on image datasets - Experiments showing how PixelCNNs can be used in conjunction with other flow layers --------------------------------------- Post rebuttal: Thank you for the rebuttal and for addressing most of the concerns. While the proposed model is very closely related to PixelCNNs and doesn't improve performance over current models in the literature, I think the paper has several strengths including the interesting connections made between autoregressive models and flows, the empirical investigation of the effect of dequantization as well as the ability to endow PixelCNNs with a form of latent space. Because of this, and after reading the other reviews and the rebuttal, I've decided to keep my score the same and I think the paper deserves to be accepted.

Strengths: Strengths: - The paper is extremely clear and well written. The motivation is clear and easy to understand and the proposed solution to the dequantization problem is simple and satisfying - The paper provides an interesting interpretation of autoregressive models as flows which, to the best of my knowledge, is novel - Thorough experiments on the effects of dequantization (both when used for training and evaluation). The experiments in section 6.2 particularly are very interesting - While other papers such as Theis et al and Ho et al have provided interesting insights on the effects of dequantization, this paper provides a nice summary of their findings and further pushes our understanding of the effects of dequantization for generative models Significance: - As the authors mention in the paper, dequantization may be the single biggest factor explaining the difference between log likelihoods of flows and discrete autoregressive models. The authors therefore tackle an important problem and open up avenues for potentially reducing this gap. Novelty: - The authors provide a novel subset flow layer and a novel interpretation PixelCNN type models. The proposed flow layer is very similar to PixelCNNs (and in fact *is* a PixelCNN for certain choices of distribution and bin conditioning), so the model itself is not particularly novel. However, the interpretation and derivation are interesting and novel on their own

Weaknesses: - One significant aspect of this problem which is not discussed in the paper is the computational cost of training these models. PixelCNNs are notoriously costly to train and the authors should provide training runtimes and a discussion of how this compares to other generative models. (see more details in question section) - It is unclear how this approach could scale to larger image datasets. As the dimension of the latent space increases it would become increasingly costly to define the mapping between hypercubes (in a similar way that PixelCNNs have a hard time scaling to a large number of autoregressive steps). (see more details in question section) - The performance of the proposed PixelFlow++ is exactly the same (2.92 bits/dim) as PixelCNN++ on CIFAR10. Is this correct? In that case, combining PixelCNNs with other flow layers doesn’t actually seem to provide any benefit.

Correctness: The claims and method are, to the best of my understanding, correct. The approach and derivations are clear and well explained and are sound. The empirical methodology is good, although it would be nice to report standard deviations over several runs (even though I know that many papers in the flow literature report a single bits/dim number instead of a mean and std dev).

Clarity: The paper is extremely well written. The language throughout is clear and concise and the model and experiments are well explained. The figures in the paper are also great and help provide a better understanding. Figure 3 in particular is excellent.

Relation to Prior Work: The relation to prior work is clearly discussed. The authors also put in a lot of effort to compare the performance of their model to a large number of baselines which is great to see.

Reproducibility: Yes

Additional Feedback: - How long does it take to train? PixelCNNs are notoriously slow to train. It would be useful to include at least a rough estimate of the training time either on single or multiple GPUs. - How does this method scale to larger datasets? All experiments in the main paper are on CIFAR10 and I suspect this is because the models are costly to train/difficult to scale. The experiments in the appendix on ImageNet32 and 64 seem to show that scaling these models seems difficult. This is of course fine, but it is a shortcoming of the model compared to other generative models and should at least be mentioned or discussed in the paper. - What do you mean by “rotations” in section 6.3? Does this simply mean changing the autoregressive order by “rotating” the feature map 90 degrees? - I may have misunderstood this, but is it necessary for the hypercubes in Z space to completely “tile” out the space? In Y space, all hypercubes fill out the [0, 256]^D space. However, it is not clear that each of the output z_upper, z_lower would correspond to hypercubes that also fill out the space. Wouldn’t this mean that the z space is disconnected? How do you do interpolations in z space then? Typo line 20: seems like there is a word missing Typo line 152: They -> they Typo line 169: dsitribution -> distribution Typo line 246: wo/ -> w/o Typo line 290: tens -> tend


Review 3

Summary and Contributions: This paper proposes a class of flows termed “subset flows,” whose purpose is to make the continuous formulation of NFs applicable to discrete data (i.e. preserve a notion of volume). Subset flows require latent-space volume to be computed efficiently, and to do this, the paper assumes the latent space is partitioned into hyper-rectangles. These hyper-rectangles are given an auto-regressive parameterization, which then leads to connections to existing autoregressive models such as the PixelCNN. The experiments demonstrate that, firstly, this perspective allows latent representations to be extracted from the PixelCNN. Secondly, the dequantization gap can be analyzed directly. Thirdly, this perspective allows PixelCNNs to be treated as a flow that’s composable with, for instance, Glow-style architectures, as the paper explores in 6.3 with good results.

Strengths: I find this paper interesting. It addresses a fundamental issue with applying flows to discrete data (i.e. how to preserve a notion of volume) and provides a novel reinterpretation of existing, popular autoregressive models such as the PixelCNN. In particular, I find its primary contributions to be: “Subset flows” and the PixelCNN: This paper defines a new type of mixture of flows (more on this below), with the clever idea of using the discrete data as the index. In consequence, this flow addresses the open problem of how to preserve a notion of volume---so that the transformation doesn’t revert into just a permutation. Cleverly, the authors choose to use the discrete data to define the partitions of a piecewise-invertible function such that the ‘pieces’ are still continuous. Showing the PixelCNN is then just one member of this broader family is an interesting and useful contribution. Exploration of the Dequantization Gap: Dequantization is a popular trick for working with discrete data, and until now, I know of no work that explicitly analyzes what is lost by its use.

Weaknesses: I find the primary weakness of the paper to be in its presentation. I believe the paper misrepresents (unintentionally) the work’s relationship to and context within the wider normalizing flows literature. There are two points that need correcting... Criticism / Characterization of Discrete Flows: The paper states that “Flow models...are naturally continuous and therefore require dequantization to be applied to discrete data” (line 39) and that “[flows] are not directly applicable to discrete data.” (lines 60): This is incorrect. As the paper states in the related work, Tran et al. [NeurIPS 2019] define discrete flows without any dequantization. In other words, the change-of-variables formula can be applied to continuous or discrete variables all the same. What the paper should say (and perhaps the authors intended something closer to this meaning) is that discrete flows are limited in practice by their inability to change volume. This is discussed at length by Papamakarios et al. [2019]; see their Section 5.3. In consequence, applying a flow to a discrete distribution can only permute the elements of the base distribution, severely limiting the expressivity of the model. “Subset Flows” as Real-and-Discrete (RAD) Flow Mixtures: The paper presents Subset Flows via the derivation used for dequantization. However, I don’t think this fully illuminates the method and its underlying principles. Rather, Subset Flows would be more accurately described as a mixture of flows---see Section 5.2 of Papamakarios et al. [2019]. Specifically, it’s a real-and-discrete (RAD) mixture, following the approach of Dinh et al. [2019], where z/y denotes the real component and the discrete index is implicitly defined by the discrete observations x. The paper should cite similar work on mixtures of flows, including Cornish et al. [2020], which uses continuous indices. References Cornish, Rob, et al. "Relaxing bijectivity constraints with continuously indexed normalising flows." ICML (2020). Dinh, Laurent, et al. "A RAD approach to deep mixture models." arXiv preprint arXiv:1903.07714 (2019). Papamakarios, George, et al. "Normalizing flows for probabilistic modeling and inference." arXiv preprint arXiv:1912.02762 (2019).

Correctness: Besides the mischaracterization of discrete flows (mentioned above), I did not find any errors of note.

Clarity: While superficially clear, the paper is misleading in its presentation of discrete flows and fails to contextualize the work within the broader normalizing flows literature---specifically, in its relationship to work on mixtures of flows.

Relation to Prior Work: No, see above.

Reproducibility: Yes

Additional Feedback: -------------------------------------------- POST-REBUTTAL UPDATE -------------------------------------------- Given the author rebuttal, I have raised my score to a 6. See below for my responses. Mischaracterization of discrete flows: Yes, I understood that you were implicitly referring to continuous flows. My issue is that the paper's statements on discrete flows are all but tautological, essentially stating 'Continuous flows are naturally continuous and therefore cannot be applied to discrete data.' It would be better to motivate the work by discussing the limitations of discrete flows---namely, no way to parameterize volume---as this is exactly what you fix! Moreover, it motivates why volume plays such a fundamental role in the work. Subset Flows as RAD Flow Mixtures: I didn't mean to imply that your work lacks novelty w.r.t. RAD---pardon me if that's how you read it. Indeed, both spaces are 'tiled' in your case whereas RAD implements a many-to-one map. Yet I am confused as to why you say "Subset flows...are themselves plain flows." Presuming 'plain flow' means continuous flow, where is the tiling structure in a (say, Glow) flow with a Gaussian base density? To me, it still seems valuable to highlight that the data's support is defining a discrete index, and in turn, the connections to previous work exploiting disjoint maps.


Review 4

Summary and Contributions: In this paper, the authors examine the extent to which normalizing flows are held back by a dequantiziation gap, where intrinsically discrete data is dequantized to a continuous distribution to be modeled by a normalizing flow. In order to remove this gap, the authors propose a new kind of normalizing flow over discrete data using continuous flows, Subset Flows, where the (finite) volume of entire subsets are tractably computable as the latent space is transformed. While the requirement is restrictive, the authors show that autoregressive models such as PixelCNN can be cast in this framework as subset flows. The new perspective as a normalizing flow enables latent space interpolations for the autoregressive models, and importantly the ability to probe the effects of dequantization both in training and evaluation. The observations here shed some light on size of the dequantization gap for normalizing flows more generally, confirming previous speculation..

Strengths: The proposed subset flows framework is an interesting theoretical contribution, even if there are few models that support tractable finite volume calculations. I feel that the real strengths of the paper are in the leveraging of this perspective to enable manipulations of the latent space for autoregressive models and the scientific contributions of empirically breaking down the dequantization gap. To my knowledge, latent space interpolations were not previously possible with autoregressive models such as PixelCNN. The authors may even want to place more emphasis on this new capability and its applications. The second component, using these autoregressive subset flows as a test bed in quantifying the extent of the dequantization gap in both training and evaluation, does not further the state of the art in modeling capability but it does shed insights on the lagging BPD of normalizing flows compared to autoregressive models. The investigation is a solid piece of science.

Weaknesses: The authors do not improve upon the performance of PixelCNN, and I think that is ok as long as it's made clear that this is not the objective. While it's great that the quantization gaps are quantified,, it would be nice if there was an outlook and followup on how these insights could be translated into improvements of standard normalizing flows. There is also the question of to what extent are the magnitude of these dequantization gaps specific to the PixelFlow model, and to what extent we should expect them to generalize to other flows. If there was some way to quantify this, that would strengthen the paper. In some ways, the work feels a bit incremental even though the investigation is thorough.

Correctness: I am confident that the claims and methodology are correct.

Clarity: The paper is very well written, however I think more details should be included in the description of bin conditioning and the interpretation of PixelCNN as a flow in the main text. As is, these sections were quite terse, requiring the reader to put things together for themselves with the information from the appendix and the workings of a regular PixelCNN. I think a little bit more elaboration and maybe a diagram would be helpful here. Perhaps algorithm 1 or 2 could be fit into the main text.

Relation to Prior Work: The paper is well situated with respect to prior work.

Reproducibility: Yes

Additional Feedback: I would put more emphasis on the capability of performing latent space interpolations and try to find a compelling application of it. -------------------------------------------- POST-REBUTTAL UPDATE -------------------------------------------- I thank the authors for the update. Although goals of this paper are somewhat modest, they are well executed and I think the paper should be accepted.

[Author Response · NeurIPS 2020]

We thank all reviewers for constructive and valuable feedback. It will be used to improve the presentation. Reviewers
note that the paper to unifies discrete autoregressive models and flows (R1,R2,R4,R5). This interpretation allow us to
study the dequantization gap, which all reviewers note as a significant contribution (R1,R2,R4,R5). Reviewers also note
that the paper is well-written and clear (R1,R2,R5). We address individual concerns below.

**Reviewer 1:**

- *Relabeling existing work:* We would like to emphasize that subset flows are mostly introduced as a vehicle to
explicitly connect flows and discrete autoregressive models. After expressing PixelCNN models as subset
flows, we are able to treat these models as building blocks in flows (which is experimented with in Sec. 6.3).
Moreover, it allows us to 1) perform latent space interpolations in PixelCNN models, which was not previously
possible and 2) explicitly quantify the effects of dequantization during training and evaluation of flow models.
- *Quadratic spline extension:* The reviewer is right that the mixture of logistics (MOL) transformation may, for
certain data, have better inductive bias than quadratic splines. One benefit of quadratic splines compared to
MOL is that it has an analytic inverse, rather than requiring the iterative bisection method.
- *Multilayer subset flows:* We again emphasize that subset flows are mostly introduced to make the explicit
connection between flows and discrete autoregressive models. As you note, extending subset flows to multiple
layers is not straightforward. We study a restricted version of multilayer subset flows in Appendix F.
- *Experiments:* The interpolation experiments are meant to demonstrate the fact that, when expressed as flows,
PixelCNN models possess a latent space. As latent space interpolation experiments have commonly been
performed for VAE-type and flow-type of models, we here demonstrate the possibility of doing this for
PixelCNN-type models, which was previously not possible. The experiments combining PixelCNN models
and coupling flows are meant to demonstrate the use of PixelCNN models as building blocks in flows. By
replacing the uniform base distribution with more flow layers, a more expressive model is obtained.

**Reviewer 2:**

- *Computational cost:* The reviewer is correct. The training of PixelCNN models is costly. In our experiments
(depending on the type of GPU used) we trained a stock PixelCNN in about 30 hours and a stock PixelCNN++
in about 10 days. We will include a comment on this in the paper.
- *Performance of PixelFlow:* You are correct that both PixelCNN++ and PixelFlow++ obtains 2.92 bpd. However,
PixelCNN++ obtains $= 2.924$, while PixelFlow++ obtains $\leq 2.917$. PixelFlow++ thus performs better than
PixelCNN++, but due to the dequantization gap and rounding, the stated number is 2.92 in both cases.
- *Rotation:* Rotations refer to rotating the feature maps $90°$.
- *Tiling:* The tiling of $\mathcal{Y}$ automatically leads to a tiling of $\mathcal{Z}$ since they are related using a bijective map.

**Reviewer 4:**

- *Mischaracterization of discrete flows:* In the statement "Flow models...are naturally continuous and therefore
require dequantization to be applied to discrete data" we are implicitly referring to continuous flows (because
the vast majority of work on flows is continuous, since, as you correctly note, discrete flows are severely limited
in their expressiveness). We agree that the statement is not true for discrete flows (as we explicitly addressed
in the related work). We will clarify that we are indeed referring to continuous flows in the statement.
- *Subset Flows as RAD Flow Mixtures:* We respectfully disagree that subset flows are better expressed as flow
mixtures. Flow mixtures involve a finite/infinite collection of flows where during sampling, the component to
use is selected before performing the transformation. RAD, for example, tiles the space $\mathcal{Y}$ and maps to all of
$\mathcal{Z}$ (not a subset of it). The transformation is therefore not invertible and during generation requires sampling
of an index that decides which region to map to. Subset flows, on the other hand, are themselves plain flows.
Subset flows tile both spaces and thus preserve invertibility. However, we are happy to include references to
these papers to discuss how this work relates to ours.

**Reviewer 5:**

- *Follow-up on insights:* This is a good point. We would suggest that there is no "discrete autoregressive models
vs. flows", but rather "how well can you reduce the dequantization gap for your flow?". Consequently, this
suggests development of methods for reducing the dequantization gap as an avenue for further research.
- *Specific to PixelCNNs:* We would not expect that the magnitude of the dequantization gaps are specific to
PixelCNN-based flows, although it is difficult to quantify the exact gap for other flow models.

**All:** We will use all comments and questions to improve the presentation. Thanks again for your work.

[Meta-Review · NeurIPS 2020]

The paper introduces a new type of flows called "subset flows" that map hypercubes to hyperrectangles, in contrast to continuous flows which map points to points. Subset flows allow tractable likelihood computation and, unlike continuous flows, can be applied to discrete data directly, without having to dequantize it first. The idea is interesting and novel, and the paper is very well written. The authors show that autoregressive models of discrete data such as PixelCNN can be interpreted as single-layer subset flows, which in addition to being an interesting insight in itself, provides a way of performing latent space interpolation with autoregressive models. All the reviewers liked the empirical exploration of the dequantization gap enabled by subset flows. The primary weakness of the paper seems to be that subset flows do not immediately lead us to models substantially different from the existing autoregressive models, though maybe this is only a matter of time (and perhaps requiring some additional insights).